# Cross-sectional study of COVID-19 knowledge, beliefs and prevention behaviours among adults in Senegal

Matthew Kearney ,[1] Marta Bornstein ,[2] Marieme Fall,[3] Roch Nianogo ,[4] Deborah Glik ,[5] Philip Massey [6]

For numbered affiliations see end of article.

**Correspondence to**
Dr Matthew Kearney;
kearnm@pennmedicine.upenn.edu

## ABSTRACT

**Objectives** The aim of the study was to explore COVID-19 beliefs and prevention behaviours in a francophone West African nation, Senegal.

**Design** This was a cross-sectional analysis of survey data collected via a multimodal observational study.

**Participants** Senegalese adults aged 18 years or older (n=1452).

**Primary and secondary outcome measures** Primary outcome measures were COVID-19 prevention behaviours. Secondary outcome measures included COVID-19 knowledge and beliefs. Univariate, bivariate and multivariate statistics were generated to describe the sample and explore potential correlations.

**Setting** Participants from Senegal were recruited online and telephonically between June and August 2020.

**Results** Mask wearing, hand washing and use of hand sanitiser were most frequently reported. Social distancing and staying at home were also reported although to a lower degree. Knowledge and perceived risk of COVID-19 were very high in general, but risk was a stronger and more influential predictor of COVID-19 prevention behaviours. Men, compared with women, had lower odds (adjusted OR (aOR)=0.59, 95% CI 0.46 to 0.75, p<0.001) of reporting prevention behaviours. Rural residents (vs urban; aOR=1.49, 95% CI 1.12 to 1.98, p=0.001) and participants with at least a high school education (vs less than high school education; aOR=1.33, 95% CI 1.01 to 1.76, p=0.006) were more likely to report COVID-19 prevention behaviours.

**Conclusions** In Senegal, we observed high compliance with recommended COVID-19 prevention behaviours among our sample of respondents, in particular for masking and personal hygiene practice. We also identified a range of psychosocial and demographic predictors for COVID-19 prevention behaviours such as knowledge and perceived risk. Stakeholders and decision makers in Senegal and across Africa can use place-based evidence like ours to address COVID-19 risk factors and intervene effectively with policies and programming. Use of both phone and online surveys enhances representation and study generalisability and should be considered in future research with hard-to-reach populations.

## INTRODUCTION

COVID-19 is an ongoing threat to global public health since its emergence at the end of 2019, yet little is known about how populations in low-resource settings have responded to COVID-19 in their daily lives. Although research is emerging about COVID-19 infection, strategies for prevention and risk factors for severe disease and mortality, there is a dearth of literature about COVID-19 beliefs, specifically in the West Africa region. For example, between January and September 2020, 4.3% of research articles about COVID-19 were focused on Africa or a specific African nation.[1] Given that ongoing prevention strategies implemented at the country, community and individual levels will be needed to curb the COVID-19 pandemic, it is critical to study health beliefs and adherence to prevention behaviours within specific social and political contexts.[2]

Providing relevant and timely research is essential for guiding evidence-based outreach and interventions, especially during public health emergencies. It is particularly

**STRENGTHS AND LIMITATIONS OF THIS STUDY**

⇒ The main strength of our study was the use of a multimodal data collection strategy, online and via telephone, which bolstered the representativeness of our total sample within Senegal and is a key strength of our study.

⇒ We adapted pre-existing research cohorts and developed novel data collection instruments to capture information about COVID-19, and this may serve as a model for the responsiveness of ongoing scholarship to future global events.

⇒ To address potential confounding between recruitment methods, we controlled for recruitment method in our multivariate regression modelling.

⇒ A study with a larger sample may have been able to identify relationships between knowledge and behaviours, and our study was not designed with enough statistical power to detect significant differences.

⇒ The findings of our study may not be generalisable beyond Senegal, or more specifically Senegalese adults.

important to advance equity within public health, and the health of populations, by understanding the social contexts and health beliefs among indigenous, marginalised and vulnerable populations around the globe.[3 4] Francophone West Africa, in particular, is under-represented in global health research because of language barriers to publication and collaboration, leading to more research in English-speaking nations like Nigeria and Ghana.[5–7]

In the context of COVID-19, researchers must learn to apply traditional research methodologies online, such as on social media, to shed light on the aetiology of COVID-19 in vulnerable and marginalised populations.[8 9] Such work creates evidence that may be applied to other geographies and enhance representation in research, which is crucial to creating tailored and effective interventions.[10] More broadly, research that creates evidence and informs policy about health issues like COVID-19 in West Africa and other parts of the developing world promotes global health equity and justice.[11–13] The current study provides timely, critical and actionable information about COVID-19 beliefs and prevention behaviours for stakeholders, policy makers and public health programmes to reduce the burden of COVID-19 in West Africa by reporting on findings among a sample of adults from the West African nation of Senegal.

## Senegal as a case study of COVID-19 beliefs and behaviours in West Africa

Senegal is the former capital of French West Africa and home to over 16 million people, and therefore may serve as a relevant case study for exploring COVID-19 beliefs in West Africa. Furthermore, Senegal's communication regarding COVID-19 has widely been considered successful and may have contributed to unexpectedly fewer COVID-19 cases and deaths.[14 15] Compared with the burden of COVID-19 in Europe, the Americas, the Middle East and South Asia, the burden of COVID-19 in Africa has been low. Relative to the rest of the world, Africa defied initial expectations by experiencing the lowest burden of reported COVID-19 cases and deaths.[16 17] Questions remain about why COVID-19 infections are reportedly low in Africa, in particular in West Africa where there have been 377 cases per 100 000 persons cumulatively. Although a 2020 analysis by Aduh and colleagues reported on perceptions and behaviours towards COVID-19 across sub-Saharan Africa, we have much to learn from populations in densely populated West Africa, a comparatively more urban sub-Saharan region home to more than 400 million people.[18 19]

Some researchers have questioned the accuracy of African COVID-19 case numbers,[16] hypothesising that the burden of disease could be much higher than reported. Uncertainty about the true trajectory of COVID-19 in Africa may leave all continents vulnerable to future outbreaks, particularly regions with low herd immunity due to lack of exposure and vaccine resource constraints. It is likely that places that have not yet needed to adopt strong prevention efforts because of reportedly low case numbers therefore may be especially vulnerable to future outbreaks. Thus, despite relatively low case numbers, it is still vital to investigate COVID-19 health beliefs and prevention behaviours in West Africa peoples to identify potential correlates and predictors of viral prevention behaviours. Our study aims to contribute evidence towards this gap in the literature by adapting pre-existing research cohorts and developing novel data collection instruments to capture information about COVID-19.

## Study objectives and research questions

To explore COVID-19 beliefs and prevention behaviours in francophone West Africa, the current study analysed cross-sectional survey data collected from a sample of adults in Senegal. The survey, which was administered both online and via telephone, included a module on COVID-19 behaviours and beliefs. We sought to identify correlates between respondent demographic attributes and COVID-19 beliefs and behaviours. Our investigation was theoretically informed from Rosenstock's Health Belief Model (HBM),[20] a behaviour change framework for public health practice positing that knowledge of a potential negative outcome and perceived threat of that outcome precede behaviour change.[21] Other recent studies have similarly relied on HBM constructs to explore COVID-19. Specifically, this study applies HBM constructs to answer the following questions:

1. In Senegal, what is the level of knowledge and awareness about COVID-19?
2. To what extent is COVID-19 perceived as a health threat?
3. What is the relationship between COVID-19 beliefs and prevention behaviours?

## METHODS
### Study design, participants and setting

From June to August 2020, we conducted a multimodal cross-sectional survey that included questions regarding COVID-19 beliefs and prevention behaviours among a convenience sample of 1452 adults in Senegal. All respondents were participating in a separate ongoing longitudinal cohort study. Potential participants were informed about the study and provided consent to participate in the survey (online) or interview (telephone). For both online and telephone participants, all individuals were required to be 18 years or older.

All online participants were part of a cohort study about entertainment education in West Africa that began in late summer 2019; of the 1286 cohort participants recruited online in summer 2019, two hundred and thirty-one both were eligible to participate (ie, Senegalese adults) and completed the COVID-19 questionnaire items. Online participants from Senegal were recruited using social media advertisements on Facebook and YouTube.

The telephone survey used a random selection of phone numbers from a pre-existing research cohort of individuals who lived in Senegal and agreed to participate

in media-related research and subsequent follow-up surveys. COVID-19 questionnaire items were included as an accompanying section with the primary study's survey. A quota system was used to match the overall population of Senegal on sex and have an even distribution of age and level of education. A total of 1221 participants were enrolled in the study via telephone, split approximately equally between participants living insider versus outside of Dakar, Senegal, and by gender.

## Data collection

This survey used two modes of data collection: online and telephone. We opted to collect data about COVID-19 from pre-existing cohorts of participants in Senegal so as to be responsive to the ongoing and evolving global epidemic and to address evidence gaps mentioned in the Introduction section. Through our multimodal strategy, responses about COVID-19 were collected from 1452 Senegalese adults. Consenting participants were compensated with 2000 Central African CFA franc (~US$4) for their time. Telephone surveys were collected by OMedia, a Dakar-based third-party research firm, and overseen by researchers from University of California Los Angeles (UCLA) (MB, DG). Online surveys were collected via Qualtrics by researchers from Drexel University (MK, PM). Incomplete responses with missing data were excluded. Figure 1 presents a flow diagram of our study recruitment and sampling.

## Measurement

Survey items included a series of self-reported demographic questions (eg, age, education) as well as a

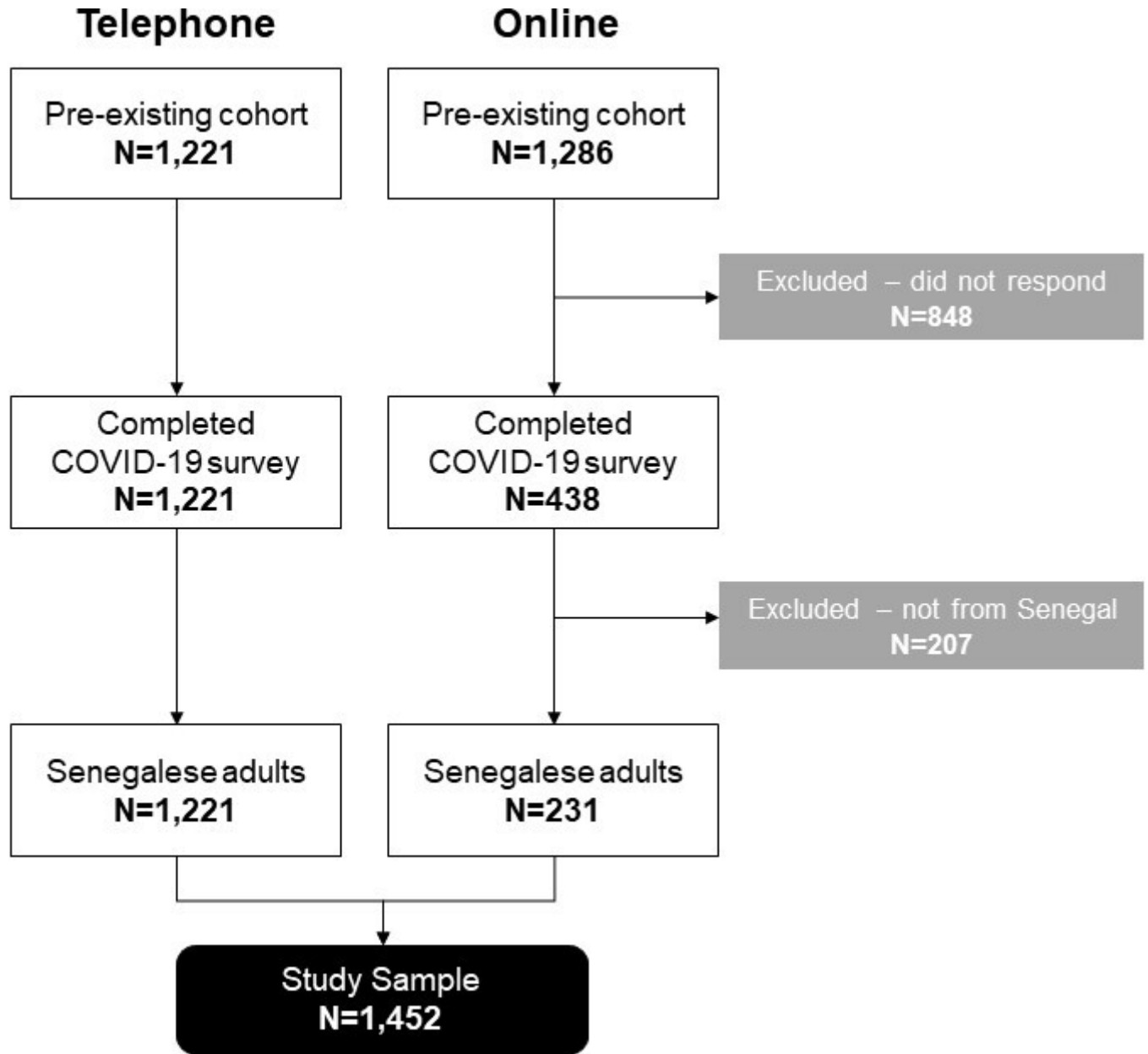

**Figure 1** Study recruitment and sampling diagram.

15-item section specifically about COVID-19. Using Cronbach's alpha, we assessed internal consistency (ie, scale reliability) for all 15 COVID-19 surveys (α=0.7; acceptable reliability). In light of the novel nature of our data collection instrument, we determined the scale to be acceptably reliable.[22] The COVID-19 section contained three measurement domains: (1) knowledge and awareness, (2) perceived threat to health, and (3) prevention behaviours. COVID-19 items were developed based on available guidance from the WHO and Centers for Disease Control and Prevention and were reviewed by French, English and Wolof-speaking team members. The survey was translated from English to French by study team members in Senegal (MF) and UCLA (RN).

Response options were binary (eg, agree/disagree) or Likert style (ie, ordinal). COVID-19 response items were aggregated into distinct scales based on the three measurement domains. Each scale ranged from 0 to 15. As the COVID-19 items were novel, these 'scales' were used as a proxy measure for determining the degree of knowledge and awareness, perceived risk and practising of known prevention behaviours, including (1) hand washing, (2) using hand sanitiser, (3) wearing a mask, (4) social distancing, and (5) staying at home. Perceived COVID-19 risk items drew from the HBM including perceived susceptibility ('Are you afraid of catching COVID-19?'; 'Do you think the COVID-19 epidemic poses a risk to Africans?'), perceived severity ('Do you think COVID-19 represents a danger to your health?'; 'How stressed do you feel about hearing about COVID-19?') and perceived behavioural control ('If you thought you had caught COVID-19, would you know who to contact?').

### Data analysis

Descriptive statistics were generated for demographic attributes and COVID-19 items. $\chi^2$ tests assessed differences in demographic attributes between the online and telephonic collection methods. Significance was determined at the α=0.05 level. For multivariate analysis, we recoded level of education, marital status, location (eg, urban/rural), religion and occupation to reduce the number of categories, creating binary or ordinal variables, as shown in table 1. Participant age was collected as a continuous measure and recoded as both quartile age groups and as binary (above vs below median age). Median values were calculated for each of the three COVID-19 scales and used as thresholds to create binary variables (eg, 0=low knowledge; 1=high knowledge).

We built five logistic regression models that predicted COVID-19 behaviours individually (ie, hand washing, mask wearing, social distancing, staying home) and in aggregate, the results of which are presented as forest plots (figure 2). For individual behaviour models, dependent variables were whether or not participants practised prevention behaviours daily in the past week; for the aggregate model, the dependent variable was the prevention behaviour scale ranging from 0 (practise no behaviours in the past week) to 15 (practise all behaviours daily). Independent variables in

| **Table 1** | Sample characteristics | | |
|---|---|---|---|
| **Characteristic** | **Category** | **n** | **%** |
| Survey modality | Telephone | 1221 | 84.1 |
| | Online | 231 | 15.9 |
| Age quartile | <21 | 353 | 24.3 |
| | 22–25 | 352 | 24.2 |
| | 26–34 | 354 | 24.4 |
| | 35+ | 393 | 27.1 |
| Sex | Female | 752 | 51.8 |
| | Male | 700 | 48.2 |
| Location | Urban | 1118 | 77.1 |
| | Rural | 291 | 20.1 |
| | Urban+rural | 42 | 2.9 |
| Education | Superior | 381 | 26.2 |
| | Secondary | 344 | 23.7 |
| | Primary | 284 | 19.6 |
| | Middle | 231 | 15.9 |
| | No | 145 | 10.0 |
| | Other | 67 | 4.6 |
| Marital status | Single | 804 | 55.4 |
| | Married—monogamous | 460 | 31.7 |
| | Married—polygamous | 161 | 11.1 |
| | Divorced | 20 | 1.4 |
| | Widower | 7 | 0.5 |
| Occupation | Informal sector | 569 | 39.2 |
| | Student | 400 | 27.6 |
| | Formal sector | 225 | 15.5 |
| | Other | 112 | 7.7 |
| | Housewife | 96 | 6.6 |
| | Retired, out of work | 49 | 3.4 |
| | Religious leader | 1 | 0.1 |
| Religion | Muslim | 1372 | 94.5 |
| | Catholic | 65 | 4.5 |
| | Protestant/Evangelical | 8 | 0.6 |
| | No | 3 | 0.2 |
| | Other | 3 | 0.2 |
| | Traditionalist | 1 | 0.1 |

n=1452.
Data collected via phone and online surveys from Senegalese adults between June and August 2020.

each model included demographic items, perceived threat (high/low) and knowledge and awareness (high/low). All statistical analyses were conducted using Stata IC software (V.15). Forest plots were generated using the 'forestplot' package in RStudio software.[23]

### Patient and public involvement

This study is a cross-sectional analysis of surveys conducted in 2020 using a multimodal recruitment strategy. No

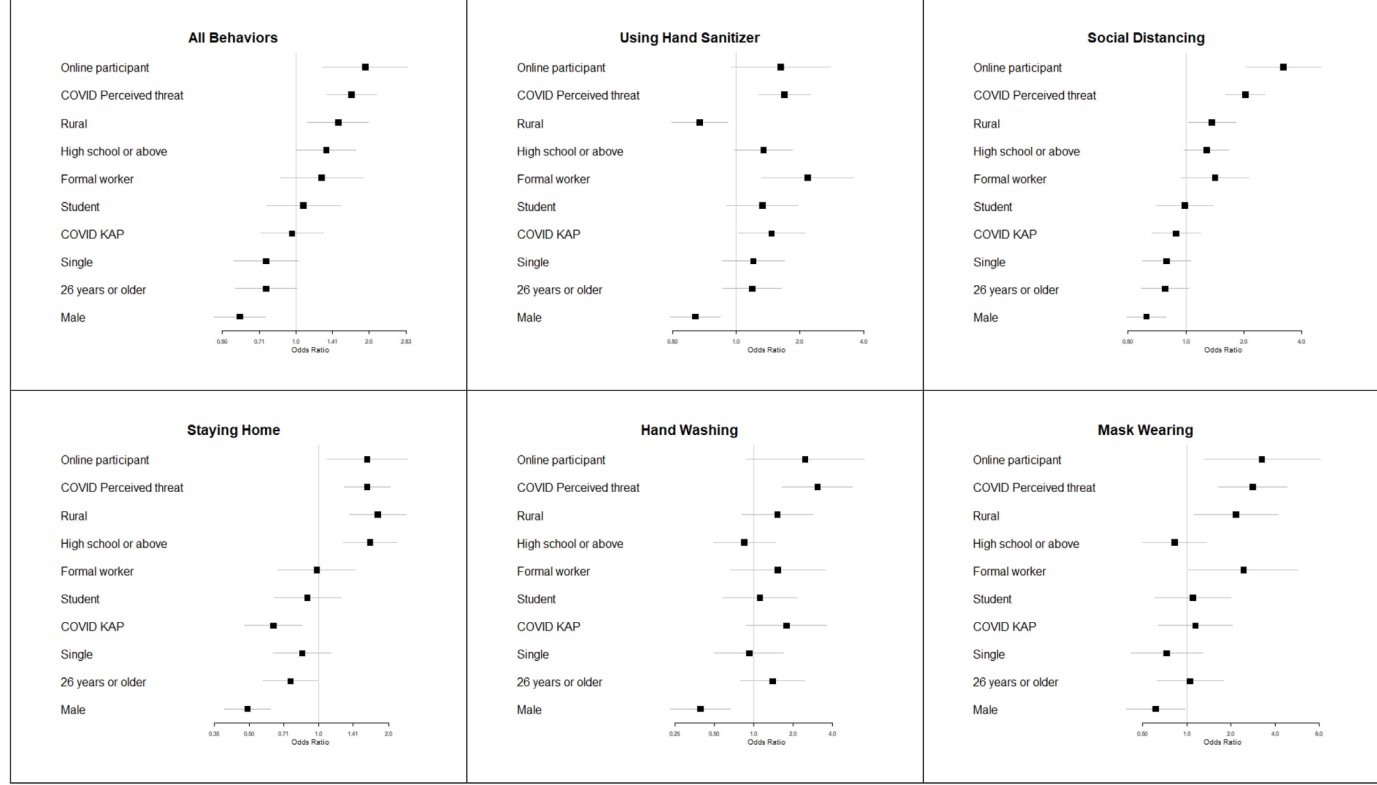

**Figure 2** Forest plot diagrams for logistic regression models predicting COVID-19 prevention behaviours (aggregate+individually). Square box represents point estimate and horizontal bar represents 95% CI. CIs that cross OR threshold of 1.0 are not statistically significant at the α=0.05 level. All estimates presented are adjusted for other model variables. n=1452. All responses collected between June and August 2020. KAP, knowledge, attitudes and practices.

patients were involved in our study. All analyses were conducted in collaboration with research partners in Dakar, Senegal, so it was not appropriate to involve specific public communities or patients in the design, conduct, reporting or dissemination plans.

## RESULTS

A total of 1452 Senegalese adults participated in our cross-sectional survey about COVID-19. Table 1 presents demographic characteristics for the study sample. Table 2 presents univariate descriptive statistics for COVID-19 survey items. Table 3 presents unadjusted regression models and table 4 presents adjusted regression models. Figure 2 presents a forest plot of our logistic regression model predicting prevention behaviours.

## Sample characteristics

Most participants identified as Muslim (94.5%; n=1372) and primarily lived in an urban area (77.1%; n=1118). Nearly half completed secondary education (ie, high school or more; 49.9%; n=725). Most participants were single (55.4%; n=804). Participants' ages range from 18 to 44 years, with a median age of 26 years and mean of 28.0 years (SD=7.6; see online supplemental file). The sample was evenly split between females and males (50.1% vs 49.9%). More than a third of participants worked in the informal sector (eg, craftsperson, restaurateur; 39.2%;

n=569) and another third were currently students (27.6%; n=400). Most participants completed the survey via phone rather than online, respectively 84.1% and 15.9%.

We identified some differences between our two recruitment methods—that is, online versus telephone (see online supplemental file). Compared with participants recruited online, participants recruited telephonically were more likely to be older on average (telephone=28.5 years vs online=25.0; p<0.001), live in rural versus urban settings (23.0% vs 4.4%; p<0.001) and have less than college education (16.1% vs 79.7%; p<0.001). Half of telephone participants were male, compared with less than 4 in 10 online participants (49.9% vs 39.4%; p=0.003). Significant differences were also identified between recruitment methods for relationship status (p<0.001), profession (p<0.001) and religion (p<0.001). Compared with telephone participants, online participants also had significantly higher COVID-19 knowledge and awareness (p<0.001) and were significantly more likely to engage in prevention behaviours (p<0.001).

## Knowledge and awareness

As shown in table 2, of the 1452 participants in our study sample less than 1% had not heard about COVID-19 (0.4%; n=6). More than 9 in 10 participants knew that hand washing (94.8%; n=1377) and mask wearing in public (97.8%; n=1420) were effective methods to prevent

**Table 2** Frequency (n) and per cent (%) for responses to COVID-19 questionnaire items

| | n | % |
|---|---|---|
| **Knowledge and awareness items** | | |
| Have you ever heard of COVID-19, also called coronavirus? | | |
| Yes. | 1446 | 99.6 |
| No. | 6 | 0.4 |
| What protective measure against COVID-19/coronavirus do you know? | | |
| Stay home. | 705 | 48.6 |
| Wash your hands regularly. | 1377 | 94.8 |
| Wear a mask in public. | 1420 | 97.8 |
| Avoid meeting with groups of more than 10 people. | 738 | 50.8 |
| **Prevention behaviour items (past week)** | | |
| Staying at home. | | |
| Partly (I try to limit time away from home but go out several times a week). | 602 | 41.5 |
| Most of the time (I go out about once a week). | 476 | 32.8 |
| All the time (I didn't leave the house). | 282 | 19.5 |
| I went out as much as usual. | 90 | 6.2 |
| Wash your hands with soap and water. | | |
| Several times a day, even if they do not seem dirty. | 830 | 57.4 |
| Every day, as soon as they are dirty after the toilet, the kitchen or the work. | 535 | 37.0 |
| Very rarely/never. | 44 | 3.0 |
| Once a day approximately. | 38 | 2.6 |
| Wear a mask when you leave your home. | | |
| Each time. | 951 | 65.6 |
| Most of the time. | 403 | 27.8 |
| Sometimes. | 90 | 6.2 |
| Never/I don't have a mask. | 6 | 0.4 |
| Use hand sanitiser gel. | | |
| Every day, but only in the places where it is needed. | 595 | 41.0 |
| Several times a day. | 528 | 36.4 |
| Rarely/never. | 215 | 14.8 |
| About once a day. | 112 | 7.7 |
| Practise social distancing (limit the number of other people). | | |
| Most of the time (I go out rarely and keep a distance). | 557 | 38.4 |
| Partly (I don't limit my social interactions but I keep my distance). | 507 | 35.0 |
| All the time (I am confined to my home). | 303 | 20.9 |
| Never (I act as usual: I pass other people and I shake their hands). | 83 | 5.7 |
| **Perceived threat** | | |
| Do you think COVID-19 represents a danger to your health? | | |

**Table 2** Continued

| | n | % |
|---|---|---|
| Yes, very serious. | 1152 | 79.6 |
| Somewhat serious. | 217 | 15.0 |
| No. | 78 | 5.4 |
| Are you afraid of catching COVID-19? | | |
| Very afraid. | 913 | 63.1 |
| Relatively afraid. | 347 | 24.0 |
| Not afraid. | 186 | 12.9 |
| Do you think the COVID-19 epidemic poses a risk to Africans? | | |
| Yes, a serious risk. | 1061 | 73.3 |
| Yes, but a reasonable risk. | 283 | 19.5 |
| No, a low risk. | 104 | 7.2 |
| If you thought you had caught COVID-19, would you know who to contact? | | |
| Yes. | 1149 | 79.4 |
| No. | 298 | 20.6 |
| How stressed do you feel about hearing about COVID-19? | | |
| Somewhat stressed. | 586 | 40.5 |
| Stressed a little. | 366 | 25.3 |
| Stressed a lot. | 262 | 18.1 |
| Not at all stressed. | 234 | 16.2 |

n=1452 Senegalese adult participants.
All responses collected between June and August 2020.

COVID-19 transmission and/or infection. About half were aware that staying at home (48.6%; n=705) or avoiding groups of more than 10 people (50.8%; n=738) were COVID-19 prevention recommendations. Nearly four in five participants (79.4%; n=1149) said they would know who to contact (eg, testing, healthcare) if they suspected themselves to have had COVID-19. The median knowledge score was 12.5 out of 15 (range=0–15; mean=11.8/SD=2.6; see online supplemental file).

### Perceived threat
As shown in table 2, 9 in 10 participants (94.6%; n=1369) perceived COVID-19 to be a very or somewhat serious danger to their health. Almost three-quarters (73.3%; n=1061) said that COVID-19 posed a serious risk to Africans. Median perceived threat score was 11.6 out of 15 (mean=10.9/SD=3.5; see online supplemental file).

### Prevention behaviours
As shown in table 2, 9 in 10 participants reported washing their hands multiple times per day (94.4%; n=1365). Nearly 8 in 10 used hand sanitiser multiple times per day (77.4%; n=1123). Approximately 7 in 10 participants wore masks every time they left their home (65.6%; n=951). Six in 10 reported practising social distancing all or most of the time (59.3%; n=1064). Half of participants reported staying home all or most of the time (52.3%; n=758). Less

**Table 3** Univariate logistic regression models predicting COVID-19 prevention behaviours (aggregate+individually)

| Predictor | All behaviours | | | Using hand sanitiser | | | Social distancing | | |
|---|---|---|---|---|---|---|---|---|---|
| | OR | 95% CI | P value | OR | 95% CI | P value | OR | 95% CI | P value |
| Online participant* | 2.39 | 1.80 to 3.18 | <0.001 | 2.46 | 1.61 to 3.75 | <0.001 | 3.33 | 2.36 to 4.70 | <0.001 |
| COVID-19 perceived threat | 1.71 | 1.37 to 2.13 | <0.001 | 1.75 | 1.35 to 2.28 | <0.001 | 2.01 | 1.62 to 2.51 | <0.001 |
| Rural† | 1.25 | 0.96 to 1.63 | 0.104 | 0.56 | 0.42 to 0.75 | <0.001 | 1.14 | 0.87 to 1.48 | 0.334 |
| High school or above‡ | 0.76 | 0.61 to 0.95 | <0.001 | 0.78 | 0.59 to 1.01 | <0.01 | 0.70 | 0.56 to 0.87 | <0.001 |
| Formal worker§ | 0.98 | 0.73 to 1.33 | 0.018 | 1.49 | 1.00 to 2.22 | 0.059 | 1.10 | 0.81 to 1.50 | 0.002 |
| Student§ | 1.56 | 1.26 to 1.94 | 0.901 | 1.54 | 1.20 to 1.98 | 0.053 | 1.67 | 1.35 to 2.06 | 0.532 |
| COVID-19 KAP | 1.47 | 1.16 to 1.86 | <0.01 | 2.13 | 1.56 to 2.90 | <0.001 | 1.55 | 1.22 to 1.97 | <0.001 |
| Single¶ | 0.92 | 0.74 to 1.14 | 0.446 | 1.07 | 0.83 to 1.37 | 0.607 | 0.96 | 0.78 to 1.19 | 0.722 |
| 26 years or older** | 0.80 | 0.64 to 0.99 | 0.044 | 1.09 | 0.85 to 1.39 | 0.511 | 0.81 | 0.66 to 1.00 | 0.048 |
| Male** | 0.49 | 0.39 to 0.61 | <0.001 | 0.60 | 0.47 to 0.77 | <0.001 | 0.52 | 0.42 to 0.64 | <0.001 |
| **Predictor** | **Staying home** | | | **Hand washing** | | | **Mask wearing** | | |
| | OR | 95% CI | P value | OR | 95% CI | P value | OR | 95% CI | P value |
| Online participant* | 1.63 | 1.22 to 2.17 | <0.01 | 2.49 | 1.07 to 5.78 | 0.029 | 2.17 | 1.04 to 4.53 | 0.036 |
| COVID-19 perceived threat | 1.63 | 1.32 to 2.01 | <0.001 | 3.86 | 2.11 to 7.05 | <0.001 | 3.16 | 1.87 to 5.34 | <0.001 |
| Rural† | 1.65 | 1.27 to 2.15 | <0.001 | 1.35 | 0.74 to 2.48 | 0.329 | 2.01 | 1.06 to 3.82 | 0.030 |
| High school or above‡ | 0.83 | 0.67 to 1.03 | <0.001 | 1.06 | 0.66 to 1.68 | 0.640 | 1.40 | 0.94 to 2.07 | 0.517 |
| Formal worker§ | 0.87 | 0.65 to 1.17 | 0.093 | 1.48 | 0.73 to 3.00 | 0.814 | 1.93 | 1.05 to 3.56 | 0.094 |
| Student§ | 1.71 | 1.39 to 2.11 | 0.348 | 0.90 | 0.58 to 1.40 | 0.273 | 0.87 | 0.58 to 1.32 | 0.035 |
| COVID-19 KAP | 0.91 | 0.73 to 1.15 | 0.439 | 2.41 | 1.29 to 4.49 | 0.005 | 1.55 | 0.93 to 2.56 | 0.090 |
| Single¶ | 1.00 | 0.81 to 1.23 | 0.985 | 0.59 | 0.37 to 0.95 | 0.028 | 0.54 | 0.35 to 0.85 | 0.006 |
| 26 years or older** | 0.74 | 0.60 to 0.91 | 0.005 | 1.53 | 0.98 to 2.41 | 0.063 | 1.39 | 0.92 to 2.11 | 0.119 |
| Male** | 0.45 | 0.37 to 0.56 | <0.001 | 0.34 | 0.21 to 0.56 | <0.001 | 0.54 | 0.35 to 0.82 | 0.004 |

n=1452.
All estimates presented are not adjusted for other model variables.
All responses collected between June and August 2020.
α=0.05.
*Ref=telephone participant.
†Ref=urban.
‡Ref=below high school.
§Ref=informal worker, housewife, retired, out of work, religious leader, other.
¶Ref=married (monogamous+polygamous), divorced, widower.
**Ref=female.
KAP, knowledge, attitudes and practices.

than 1 in 10 people went out as usual (6.2%; n=90) or ignored social distancing (5.7%; n=83). Less than 1 in 100 participants never wore a mask in public (0.4%; n=6). The median prevention behaviours score was 11 out of 15 (mean=10.5/SD=2.3; see online supplemental file). Unadjusted bivariate regression models for predictors of prevention behaviours are presented in table 3 for reference. Below we discuss the results for adjusted regression models shown in table 4 and figure 2.

### Regression analysis
COVID-19 behaviours were significantly associated with COVID-19 beliefs and participant demographic attributes (see figure 2, table 4). We also observed differences in behaviours based on our participant recruitment method. Compared with participants recruited telephonically,

participants recruited online were significantly more likely to practise social distancing (OR=3.33; p=<0.001), wear masks (OR=3.26; p=0.007) and stay home (OR=1.63; p=0.013).

Participants who reported high degrees of perceived threat about COVID-19 were significantly more likely to practise prevention behaviours in both the aggregate and individual models (OR=1.69; p<0.001). Knowledge and awareness related to COVID-19 poorly predicted practising prevention behaviours, although high levels of knowledge and awareness were positively associated with participants' use of hand sanitiser (OR=1.47; p=0.035). At the same time, higher levels of knowledge and awareness about COVID-19 were negatively associated with staying home (OR=0.64; p=0.002).

**Table 4** Multivariate logistic regression models predicting COVID-19 prevention behaviours (aggregate+individually)

| | All behaviours | | | Using hand sanitiser | | | Social distancing | | |
|---|---|---|---|---|---|---|---|---|---|
| **Predictor** | **aOR** | **95% CI** | **P value** | **aOR** | **95% CI** | **P value** | **aOR** | **95% CI** | **P value** |
| Online participant* | 1.93 | 1.29 to 2.87 | 0.001 | 1.62 | 0.94 to 2.78 | 0.079 | 3.2 | 2.05 to 5.00 | <0.001 |
| COVID-19 perceived threat | 1.69 | 1.33 to 2.13 | <0.001 | 1.69 | 1.28 to 2.22 | <0.001 | 2.03 | 1.61 to 2.57 | <0.001 |
| Rural† | 1.49 | 1.12 to 1.98 | 0.006 | 0.67 | 0.5 to 0.91 | 0.01 | 1.36 | 1.03 to 1.81 | 0.032 |
| High school or above‡ | 1.33 | 1.01 to 1.76 | 0.044 | 1.35 | 0.98 to 1.84 | 0.062 | 1.28 | 0.98 to 1.67 | 0.07 |
| Formal worker§ | 1.28 | 0.86 to 1.9 | 0.23 | 2.18 | 1.32 to 3.6 | 0.002 | 1.42 | 0.95 to 2.13 | 0.09 |
| Student§ | 1.07 | 0.76 to 1.52 | 0.691 | 1.33 | 0.9 to 1.97 | 0.149 | 0.98 | 0.7 to 1.38 | 0.927 |
| COVID-19 KAP | 0.96 | 0.72 to 1.29 | 0.798 | 1.47 | 1.03 to 2.1 | 0.035 | 0.89 | 0.66 to 1.18 | 0.411 |
| Single¶ | 0.75 | 0.56 to 0.56 | 0.07 | 1.2 | 0.86 to 1.69 | 0.281 | 0.79 | 0.59 to 1.06 | 0.116 |
| 26 years or older** | 0.75 | 0.56 to 1.00 | 0.054 | 1.19 | 0.86 to 1.64 | 0.287 | 0.78 | 0.59 to 1.03 | 0.077 |
| Male** | 0.59 | 0.46 to 0.75 | <0.001 | 0.64 | 0.49 to 0.84 | 0.001 | 0.62 | 0.49 to 0.79 | <0.001 |
| | **Staying home** | | | **Hand washing** | | | **Mask wearing** | | |
| **Predictor** | **aOR** | **95% CI** | **P value** | **aOR** | **95% CI** | **P value** | **aOR** | **95% CI** | **P value** |
| Online participant* | 1.62 | 1.09 to 2.42 | 0.018 | 2.48 | 0.89 to 6.96 | 0.084 | 3.25 | 1.31 to 8.04 | 0.011 |
| COVID-19 perceived threat | 1.62 | 1.29 to 2.03 | <0.001 | 3.09 | 1.67 to 5.71 | <0.001 | 2.81 | 1.64 to 4.82 | <0.001 |
| Rural† | 1.8 | 1.35 to 2.38 | <0.001 | 1.52 | 0.81 to 2.86 | 0.189 | 2.17 | 1.12 to 4.19 | 0.021 |
| High school or above‡ | 1.67 | 1.28 to 2.17 | <0.001 | 0.85 | 0.49 to 1.45 | 0.546 | 0.82 | 0.5 to 1.36 | 0.448 |
| Formal worker§ | 0.98 | 0.67 to 1.44 | 0.918 | 1.54 | 0.67 to 3.53 | 0.311 | 2.44 | 1.03 to 5.75 | 0.042 |
| Student§ | 0.89 | 0.64 to 1.25 | 0.51 | 1.12 | 0.58 to 2.16 | 0.733 | 1.1 | 0.6 to 1.99 | 0.76 |
| COVID-19 KAP | 0.64 | 0.48 to 0.85 | 0.002 | 1.78 | 0.88 to 3.63 | 0.11 | 1.14 | 0.64 to 2.04 | 0.654 |
| Single¶ | 0.85 | 0.64 to 1.13 | 0.26 | 0.93 | 0.5 to 1.71 | 0.809 | 0.73 | 0.41 to 1.29 | 0.277 |
| 26 years or older** | 0.76 | 0.58 to 0.99 | 0.045 | 1.4 | 0.8 to 2.46 | 0.243 | 1.05 | 0.63 to 1.77 | 0.847 |
| Male** | 0.49 | 0.39 to 0.62 | <0.001 | 0.39 | 0.23 to 0.66 | <0.001 | 0.61 | 0.39 to 0.97 | 0.037 |

n=1452.
All estimates presented are adjusted for other model variables.
All responses collected between June and August 2020.
α=0.05
*Ref=telephone participant.
†Ref=urban.
‡Ref=below high school.
§Ref=informal worker, housewife, retired, out of work, religious leader, other.
¶Ref=married (monogamous+polygamous), divorced, widower.
**Ref=female.
aOR, adjusted OR; KAP, knowledge, attitudes and practices.

Compared with participants with less than a high school degree, participants with high school education or above were more likely to stay home (OR=1.67; p<0.001), practise social distancing (OR=1.28; p=0.070) and use hand sanitiser (OR=1.35; p=0.062). Compared with urban participants, rural participants were more likely to practise social distancing (OR=1.36; p=0.032), stay home (OR=1.80; p<0.001) and wear a mask (OR=2.17; p=0.021), but less likely to user hand sanitiser (OR=0.67; p=0.010). Use of hand sanitiser was more likely among formal workers compared with students (OR=2.18; p=0.002). Compared with participants less than 26 years old, older participants (over 26) were significantly less likely to stay home (OR=0.76; p=0.045).

Sex-based differences were observed for COVID-19 prevention behaviours. Males had 41% lower odds of practising *all five* COVID-19 prevention behaviours compared with females (OR=0.59; p<0.001). Males also reported lower adherence to COVID-19 prevention measures for each of the five behaviours individually. Compared with females, males were 61% less likely to wash their hands on a daily basis (p<0.001), 51% less likely to report staying home (p<0.001), 39% less likely to wear masks daily (p<0.037), 38% less likely to practise social distancing (p<0.001) and 37% less likely to use hand sanitiser (p=0.001).

## DISCUSSION

In our cross-sectional sample of Senegalese adults, we observed that the majority of participants practised COVID-19 prevention behaviours on a daily basis. Mask

wearing, hand washing and use of hand sanitiser were most frequently reported. Social distancing and staying at home were also reported although to a lower degree. We also identified a range of psychosocial and demographic predictors for COVID-19 prevention behaviours. For example, participants who perceived COVID-19 to be a threat were significantly more likely to practise every prevention behaviour measured than those who did not perceive COVID-19 to be a threat. This is in line with existing health behaviour research regarding perceived threat as a necessary precursor to behaviour change.[20 21] Our findings from Senegal contribute to a growing body of literature about COVID-19 beliefs and behaviours in developing African nations and support prior research in both developing and developed nations about predictors of health behaviours.

### Prevention behaviours

Males were less likely than females to report practising any of the prevention behaviours that we studied. In the USA, a developed nation, previous research shows that females are more likely to engage generally in preventive health behaviours adhering to public health recommendations, such as cancer screenings, exercise and dietary intake.[24 25] Specific to COVID-19, females have also been shown to more frequently practise prevention behaviours like mask wearing, and this is true both in developed nations like the USA,[26 27] as well as in developing nations like South Africa,[28] Chile,[29] Turkey[30] and China.[31] However, studies in Sierra Leone and Cameroon have both found that females were less likely to engage in COVID-19 prevention behaviours, contradictory to our findings from Senegal. In general, males may also be more likely to engage in risk-taking behaviours[32] whereas females are more risk averse.[33] Future research into adherence of recommended COVID-19 prevention behaviours should explore sex-based differences in risk perceptions and tolerance.

Practising COVID-19 prevention behaviours was also positively associated with living in a rural versus urban area. It is possible that people residing in rural areas have an easier time implementing preventive behaviours such as staying at home and social distancing than people in more densely populated urban areas. Finally, participants with at least a high school education were also more likely to engage in preventive behaviours. Educational differences in adoption of COVID-19 preventive health behaviours have previously been reported in studies in other nations like South Africa and China.[31 34 35] Such demographic variations in practising COVID-19 prevention behaviours indicate priority populations for future health communication campaigns. For example, people in urban areas may require more prompting and resources to implement preventive behaviours in the future.

Considering our multimodal recruitment strategy, we found that respondents from the online sample demonstrated greater knowledge and awareness of COVID-19 and engaged in prevention behaviours more often compared with respondents in the telephone sample, suggesting that strategies leveraging online communications and communities of users will be important to continue supporting and promoting prevention behaviours, in particular to build vaccine confidence as vaccine availability increases throughout the region. Differences between respondents based on recruitment method could indicate similar differences in Senegalese adults in general depending on whether they had access to and were frequent users of the internet. This provides support for both digital and on-the-ground messaging and communication to strengthen pro-prevention beliefs, attitudes and behaviours. For example, health promotion campaigns could engage local social media influencers or bloggers to promote positive attitudes towards COVID-19 prevention behaviours, including vaccination, and rely on their networks of users to further disseminate health communications via word of mouth to their families and friends online and offline.

### Health beliefs

Our study findings comport with a robust body of evidence about COVID-19 knowledge, awareness, attitudes and practices. In Senegal, we observed that knowledge, perceived risk and practising prevention behaviours for COVID-19 were high in general—although with some demographic variations. A 2020 international review by Puspitasari and colleagues reported generally high knowledge and awareness about COVID-19, pro-prevention attitudes and high adherence to COVID-19 prevention behaviours like hand washing and social distancing.[36] Closer to the West African and Senegalese contexts, Aduh and colleagues reported moderate to high levels of COVID-19 knowledge and perceived risk in sub-Saharan Africa.[18]

We did not find an association between knowledge and engaging in prevention behaviours, a finding both contrary to and in line with prior international research of infectious disease practices in developing nations. For example, a 2020 study in Guyana found that knowledge was a significant predictor of practising malaria and dengue prevention behaviours but not for Zika virus and leishmaniosis, a common protozoan-caused skin infection.[37] Specific to COVID-19, a 2020 study in China reported that participants with increased knowledge were more likely to go to crowded places and less likely to wear masks.[31] Finally, a 2021 scoping review of 28 studies on COVID-19 beliefs and preventive practices in nine sub-Saharan African nations—not including Senegal—reported that although knowledge about COVID-19 was generally high, it was not always associated with increased risk perceptions or attitudes towards the virus, and by extension was a poor predictor of prevention behaviours.[38]

In the current study, we found that perceived COVID-19 risk was a stronger predictor of prevention behaviours, which is also supported by previous international research.[2 39] Low perceptions of risk may result in low compliance with virus restrictions and adherence

to prevention behaviours,[40–42] and therefore facilitate COVID-19 spread. This confirms another recent study which recruited using social media from the USA, China, Taiwan and Mexico.[2] Hsing and colleagues found that perceived COVID-19 severity and susceptibility—constructs that comprise perceived risk/threat in the HBM—were both predictors of hand washing and social distancing.[2] In our study based in Senegal, perceived risk predicted hand washing and social distancing, and predicted staying home, using hand sanitiser and mask wearing. These findings suggest that to prevent COVID-19 spread in Senegal and other parts of francophone West Africa, we need to focus more on perceived risk than knowledge. For example, health agencies and organisations should promote health communications that highlight Senegalese residents' susceptibility to contracting COVID-19 as well as its potential health impacts. Messaging should be targeted towards groups that we identified as having lower adherence to prevention behaviours, in particular men.

## A focus on Senegal

In general, Senegalese participants had high levels of COVID-19 knowledge and awareness, perceived COVID-19 to be a threat both personally and across Senegal and, in most cases, adopted known prevention behaviours on a daily or regular basis. Looking forward, as vaccines become more widely available in Senegal and many West African nations, inequities in access will make sustaining preventive behaviours paramount to continued control of the virus.[43] Our findings suggest that other nations should look to Senegalese communications about COVID-19 to learn messages that may resonate more effectively with their populations to increase prevention behaviours.

As has been stated, most participants reported moderate to high adherence to COVID-19 prevention behaviours. To explain our findings, we must consider lessons learnt from previous regional epidemics and how they relate to evolving social contexts (eg, norms, culture, population attributes), in particular Ebola.[9] Many behaviours associated with preventing COVID-19 were already familiar to residents of African nations who experienced Ebola outbreaks, including hand washing, use of personal protective equipment and social distancing.[44] COVID-19 in West Africa comes several years after the Ebola outbreak in 2014–2015, which tested the limits of the health systems and demonstrated its resilience. For example, although Senegal experiences a nationwide shortage of doctors—just 7 per 100 000 persons in 2020—the nation has been comparatively effective at controlling COVID-19 infections. The first case of COVID-19 in Senegal, a country of 17 million, was reported in early March 2020, prompting the Ministry of Health to develop protocols to monitor and address cases within Senegal and at its borders and airports. Cases in Senegal remained low throughout spring, but like other nations rose considerably in the latter half of 2020 and first months of 2021.[45 46] On 8 August 2020, Senegal's Minister of Interior mandated mask wearing in indoor spaces, banned gatherings in public spaces like beaches, theatres and sporting grounds and prohibited demonstrations on public highways.[47] These restrictions were emphasised in particular for the densely populated region of Dakar, but did not include mandatory lockdowns, local travel restrictions or school/university closures. Although our survey was conducted in summer 2020, our findings continue to provide relevant context to current COVID-19 epidemiological data in Senegal: as of 27 January 2022, the 7-day rolling average for COVID-19 cases per million people was lower in Senegal than both the rest of Africa and the world—7.95 vs 22.65 vs 427.1, respectively.[48]

Senegal's success at controlling the virus throughout 2020 has been attributed to many factors, including a citizenry who has witnessed the devastating impacts of the Ebola epidemic. More broadly across West Africa, citizens may be more likely to comply with public health restrictions because of their experience with Ebola,[49] and this extends to individual behaviour changes as well. For example, one study reported that hand washing was shown to increase in Senegal as a result of Ebola education campaigns.[50] In our analysis, we found that daily hand washing was the prevention behaviour most often reported by participants. Future health campaigns may look at Ebola prevention messages to identify behaviours and information that may be reinforced in the context of COVID-19.

## Limitations

Our findings may not be generalisable beyond Senegal, or more specifically Senegalese adults, because of our targeted multimodal recruitment strategy. The recruitment strategy increased the potential for selection bias because participants were recruited online and via telephone in Senegal; thus, all participants, by the nature of the recruitment methods, had access to the internet and/or a phone. We did not identify significant relationships for many factors in our regression model because our cross-sectional study was not statistically powered to detect significant differences and we did not assess the discriminatory power of our novel questionnaire items or scales. Staying at home or social distancing prevention behaviours may have been confounded by other factors not included in our regression modelling, in particular government lockdowns, local travel restrictions, school/university closures and whether a participant worked outside of the home or not. Although our COVID-19 survey items were developed by the study team based on available WHO guidance, they were not validated tools like the International Citizens Project COVID-19 survey and therefore have limited transferability to similar investigations.[28]

## CONCLUSION

Our study contributes to a growing body of literature about COVID-19 in Africa,[28 51 52] and is the first study to look specifically at knowledge and prevention behaviours

in the francophone West African nation of Senegal. In Senegal, we observed that knowledge and perceived risk of COVID-19 were high in general. But, risk was a stronger and more influential predictor of COVID-19 prevention behaviours. COVID-19 prevention behaviours also varied by sex, education, location and country. These findings suggest that in West Africa, nations should seek to elevate risk perceptions about COVID-19 and develop messaging to better target certain groups. Elevating the perceived risk of a population is a difficult issue because public health programmes do not want to raise unnecessary alarm. Thus, messages around risk of COVID-19 should be accompanied by empowering messages of how individuals can protect themselves, their families and their communities. Health campaigns should look backwards to successes with Ebola for improving health messaging about COVID-19, and may also look to replicate the successes of regional examples like Senegal.

**Author affiliations**
[1]Department Family Medicine and Community Health, University of Pennsylvania Perelman School of Medicine, Philadelphia, Pennsylvania, USA
[2]Department of Epidemiology, Ohio State University College of Public Health, Columbus, Ohio, USA
[3]The African Health and Education Network (NGO RAES), Dakar, Senegal
[4]Department of Epidemiology, UCLA Fielding School of Public Health, Los Angeles, California, USA
[5]Department of Community Health Sciences, UCLA Fielding School of Public Health, Los Angeles, California, USA
[6]Department of Health, Human Performance and Recreation, University of Arkansas College of Education and Health Professions, Fayetteville, Arkansas, USA

**Acknowledgements** We are grateful for support from our French and Senegalese collaborators in Dakar, Senegal, at The African Health and Education Network (NGO RAES) and at Université Cheikh Anta Diop. Data collection was completed while the first author (MK) and the last author (PM) were affiliated with the Drexel University Dornsife School of Public Health, and while the second author (MB) was a graduate student at UCLA Fielding School of Public Health. Finally, we are thankful to all participants in this study.

**Contributors** All authors contributed to the editing and reviewing of this paper. MK, MB, DG and PM were involved in conceptualising this article. MF and RN led the development of the data collection tool. MK and RN designed the analyses for this manuscript, and MK conducted the analyses. MK, MB, DG and PM wrote the paper. All authors reviewed, edited and revised the paper. MK acts as guarantor for the final manuscript.

**Funding** This study received funding from the Bill & Melinda Gates Foundation (OPP1181104).

**Disclaimer** The study sponsors had no role in study design, data collection, data analysis, our decision to publish or preparing the manuscript.

**Competing interests** None declared.

**Patient and public involvement** Patients and/or the public were not involved in the design, or conduct, or reporting, or dissemination plans of this research.

**Patient consent for publication** Not required.

**Ethics approval** All study procedures and protocols were approved by the institutional ethical review board at Drexel University (protocol ID: 00015164) and UCLA (protocol ID: 16-000393). Written informed consent was obtained from all participants.

**Provenance and peer review** Not commissioned; externally peer reviewed.

**Data availability statement** Data are available on reasonable request

**ORCID iDs**
Matthew Kearney http://orcid.org/0000-0003-2898-573X
Marta Bornstein http://orcid.org/0000-0002-5200-8181
Roch Nianogo http://orcid.org/0000-0001-5932-6169
Deborah Glik http://orcid.org/0000-0002-2532-3661
Philip Massey http://orcid.org/0000-0002-0577-8618

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
