## [Reviewer comments · BMJ Open]

ARTICLE DETAILS

TITLE (PROVISIONAL)	A cross-sectional study of COVID-19 knowledge, beliefs and prevention behaviors among adults in Senegal
AUTHORS	Kearney, Matthew; Bornstein, Marta; Fall, Marieme; Nianogo, Roch; Glik, D; Massey, Philip

VERSION 1 – REVIEW

REVIEWER	Wong, Cho Lee Chinese Univ Hong Kong, The Nethersole School of Nursing
REVIEW RETURNED	10-Nov-2021

GENERAL COMMENTS	The current study is a cross-sectional study aimed at exploring the knowledge, beliefs and prevention behaviors of COVID-19 among adults in Senegal. As the author mentioned, little is known about how people in low-resource settings responded to COVID-19. Therefore, current study represented an important topic to explore COVID-19 behaviors and beliefs in West Africa. However, I have some concerns about the current manuscript, the details are as follows. Major concerns 1. Two recruitment methods including online and telephone were used. The samples recruited by telephone were from a panel of phone numbers of individuals who agreed to participate in media-related research, but the samples recruited online was part of a cohort study on entertainment education in West Africa. There were obvious selection bias that might affect the results.2. Throughout the paper, the introduction part is not directly related to the research purpose, various aspects of the methodology seem to be unclear, and some concern was raised about the statements in the discussion part beyond the available results. Other comments Introduction 3. The authors described quite a lot of West African context (lines 6-44, page 6), but it seems not directly related to the research purpose.4. Line 27, page 8, a repeated “that”5. Line 54, page 8. The authors described “this study applies constructs from the Health Belief Model to answer the following questions”. However, how Health Belief Model guide the study should be described in more detail. Method 6. The sample size calculation should be presented.7. Since the measurement was self-developed based on the HBM constructs, please indicate which item belongs to which HBM construct. Please also present the Cronbach’s alpha of this scale in this study. Besides, the measurement included three scales of
--

	knowledge, belief and behavior, and the total items was only 15. It was questionable whether the measurement was comprehensive enough. 8. For data analysis, (1) the normality of the data may needed to be tested and presented; (2) “Pearson’s correlation coefficient to explore relationships between items in the COVID-19 scales” (Lines 21-23, page 11) needed to be further clarified, as this measurement included three scales and 15 items. It was not unclear which relationship between them has been explored; (3) please clarify whether all demographic items, perceived threat, and knowledge & awareness were included in the model of regression analysis or only the ones that showed significance in bivariate analysis (regarded as potential factors of behavior) were included in the model; (4) the significant level may need to be presented. Results 9. The authors compared the differences between the participants recruited online and via telephone. What did it have to do with the research purpose? In addition, as the samples of the two recruitment methods were not from the same population group, the significant difference between the two recruitment methods might not be caused by the recruitment method but the samples themselves. 10. The behavior of staying at home or social distancing may be affected by various factors such as government management measures and the working status of the participants. Therefore, it might not totally reflect the prevention behavior of someone with or without work. Discussion 11. The suggestion that “online communication as an important strategy to build vaccine confidence.....” seems cannot conclude directly from the result. 12. The authors did not find an association between knowledge and prevention behaviors. Please elaborate on whether this result was consistent with previous studies. Please clarify whether the knowledge scale has sufficient discriminative power to detect the differences considering the difficulty as well as the small number of the items. 13. The implication of the research regarding the significant results on the following COVID-19 prevention needs further discussion that focuses more on the result of the current study.
--	---

REVIEWER	Lalla-Edward, Samanta Tresha University of the Witwatersrand Faculty of Health Sciences, Ezintsha, a division of Wits Health Consortium
REVIEW RETURNED	12-Nov-2021

GENERAL COMMENTS	bmjopen-2021-057914 A cross-sectional study of COVID-19 knowledge, beliefs, and prevention behaviors among adults in Senegal Thank you for the opportunity to review your manuscript. The review may seem harsh however if you can address the comments I feel that it would significantly improve the quality and readability of your manuscript and place it in bed stead for publication and citing.
--

Title: Please consider revising to include a digital health focus (if apt or you consider the comments hereabout)

Abstract:

Results: You do not mention the total number of people in the study

Conclusion is not related to the results.

Keywords: These are meant to increase the chances of your article coming up in a search therefore these should not be a repeat of words in your title. Also, there is something wrong with your keywords a from formatting perspective. Consider corona, SARS COV-2, French West Africa, something digital health if you change the focus

Major comments:

Although not necessarily published by others, I question the relevance of your results to present times/current behaviour. You are attempting to publish your findings over a year after you collected the data. In the case of COVID-19 much has rapidly changed since it's emergence and what people knew then (at the time when you collected the information) relative to what they know now may /should be quite different. This means their behaviours could be different and your recommendations less valid/useful.

The objectives of your study are focussed on quantifying knowledge, awareness, and perceived threat and determining/describing relationships between beliefs and behaviours. You used online and telephone recruitment and data collection because these were allowable methods in the pandemic and not because of its merit. If there was no pandemic this data collection would have been done in person. However, in several places in the manuscript you report on, and promote the digital data collection – as if it was something you set out to do originally. Therefore, I suggest you consider how you want to incorporate your data collection methods into this paper and reword the paper such that it becomes appropriate. So, if you want to report on this and make conclusions and recommendations then you need to be explicit that it was not a focus of the study but when you analysed the data you found XXX and it was worthy of noting.

Be careful about making sweeping statements about novelty of the work and data collection tools and recommendations its future use; a bit of this work has been done in Africa and globally. I didn't see your complete tool (if different from what was reported) but based on the tables in the results I can say that while your tools may be novel, the Likert scales particularly seem a bit strange and there are more informative (like used in ICP Covid) and even validated tools available for implementation.

Article summary:

It would be better to organise the strengths together and the weaknesses together.
While I can understand the point you are making in the first two bullets it does seem like a bit of a contradiction. On the one hand you list your method as preventing selection bias and on the other you talk about it increasing selection bias. A similar comment about generalisability. You report that what you did increases representativeness and generalisability but also list a limitation about the findings possibly only being applicable to Senegalese adults.

Introduction:

The introduction is long and contains unnecessary descriptions about West Africa. You even talk about mobile technology penetration here as if intentional to the study.
What I think is missing and would have been very useful to contextualising the results particularly, is if you described what the COVID-19 regulations and restrictions were at the time of conducting the study. I feel without this it leaves the results open to much interpretation. For e.g.,
you report on males being more likely to not remain at home, social distance etc. so the recommendation would be to prioritise men for COVID-19 prevention messaging/behaviour change. However, if the country was not in lockdown and people were still able to travel (locally) without restriction and were going to work as normal then I would say that men may have been doing what they were always doing and are not a priority population.
If schools /universities were closed and considering that 1/3 of your participants were students, then it would stand to reason that those with a high school education or above were more likely to stay at home. If schools were open – this may not be the results.

Methods:

The primary study and your secondary data are very unclear. You mention secondary data analysis in the abstract, but in the methods it seems like you did a sub-study or used convenience sampling to recruit people and this was not secondary data analysis. I would suggest
Clearly describe the primary study from where you extracted data to do this secondary data analysis
Include recruitment flow diagram which shows who and how many were approached/screened, excluded (with reasons), and finally included in the study and analysis. This helps the reader to understand your processes.
It would also be useful to elaborate on the HBM and what you specifically focussed on from the model for your design/assumptions/guidance for the study and questionnaire design.
Where is Appendix A?
With the questionnaire – were there any pilots? How did you know that you asked the correct questions? Who did the data collection?
The sample size is also not clear. Since you mentioned that the sample was not adequately powered, I assume that this was a convenience sample of X or get what you get? If not – what informed when you stopped recruiting/collecting data?

Results:

Overall, the readability of this section is low. The way data is presented in the text and tables and the flow make it a bit difficult to follow.

You could give your sections headings like COVID-19 level of knowledge and awareness, perceived threat, etc. instead of the research questions.

Consider presenting the results just as n (%) or vice versa. Three people in 10, 20 people in 100 to represent 30% and 20% is a non-conventional way to report these results and is also unnecessary. Just report as the data as is.

What is the value add of having both Figure 1 and Table 3?

I think for a clearer presentation it would have been nice to see your sample characteristics/demographics (as you have presented), another table with correlations between the characteristics/demographics and the outcome variables and then the logistic regression. Currently Table 3 is not easy to read, and one must go back and forth with the other tables to check the categories for each characteristic. Is it not possible to at least have all the data in a supplementary table if this summary regression table is to be included in the manuscript?

If accepted I assume the editorial team will work with you to revise your Table legends

Discussion:

In the first part of the discussion, you lack contextualising your findings. You just either include references next to statements or say previously reported. Are these African studies, many different countries/ globally? High income/low income etc.? Why is this important – it allows you to make more meaningful conclusions about how your results compare.

Our results suggest that online communication will be an important strategy to build vaccine confidence and to continue supporting prevention behaviors especially as vaccine availability increases throughout the region. Respondents from the online sample demonstrated greater knowledge and awareness of COVID-19 and engaged in prevention behaviors more often compared to respondents in the telephone sample. This provides support for both online and on-the-ground messaging and communication to strengthen pro-prevention beliefs, attitudes, and behavior.

What is the point of this paragraph? First you say online is the way to go then you say online and, on the ground, need to be done.

What does this mean? This may serve as a model for the responsiveness of ongoing scholarship to future global events. Avoid repeating the results excessively in your discussion and conclusion – we read it in the results section, rather tell us the so what/implications.

Conclusion:

The first paragraph seems less relevant to your study and if omitted would make no difference.

	References: There are several references that you could look at to improve the discussion and reference list including: Gholizadeh et al, 2021 Health Promotion Perspectives Majam et al, 2021 PLoS One Salyer et al, 2021 Lancet U Kollamparambil et al, 2021 PLoS One Unicef 2021 Jaja et al, 2020 Emerging Microbes and Infections Assefa et al, 2021 American Journal of Tropical Medicine Formatting/Editing: Re-read and correct for tense and grammar throughout the manuscript e.g., design (article summary L45), insider (data collection L 43), ranged (sample characteristics L47) etc. You write the methods and results particularly in mixed tense – past and future. As the study has already occurred you should mainly write in the past. Use sex and gender interchangeably. You are writing about sex as far as I can see – edit appropriately throughout. There are several instances of repetition e.g., telephone, online, study taking place in Senegal, francophone references. Please try and remove these. Avoid stating the obvious – e.g., more men versus women – if you say more men, implicit is the comparator being women, most participants completed the survey via phone rather than online, respectively 84.1% and 15.9%. You reported earlier telephone=1,221; online = 231 so more phone than online is already known. Good luck with the revisions! 1
--	--

VERSION 1 – AUTHOR RESPONSE

Reviewer: 1

Dr. Cho Lee Wong, Chinese Univ Hong Kong Comments to the Author:

The current study is a cross-sectional study aimed at exploring the knowledge, beliefs and prevention behaviors of COVID-19 among adults in Senegal. As the author mentioned, little is known about how people in low-resource settings responded to COVID-19. Therefore, current study represented an important topic to explore COVID-19 behaviors and beliefs in West Africa. However, I have some concerns about the current manuscript, the details are as follows.

Major concerns

1. Two recruitment methods including online and telephone were used. The samples recruited by telephone were from a panel of phone numbers of individuals who agreed to participate in media-related research, but the samples recruited online was part of a cohort study on entertainment education in West Africa. There were obvious selection bias that might affect the results.

Thank you for your feedback on potential selection bias. We agree that our recruitment strategies increased selection bias that may have affected the results, and address this in our article summary, results and limitations sections. During our analysis, we also controlled for recruitment method to address potential confounding.

Article Summary: “Our recruitment strategy may have also increased the potential for selection bias because participants were recruited online and on-the-ground in Senegal; thus, all participants, by the nature of the recruitment methods, had access to the internet and/or a cell phone. To address potential confounding between recruitment methods, we controlled for recruitment method in our multivariate regression modelling.”

Results: “We identified some differences between our two recruitment methods – i.e., online versus telephone (data not presented). Compared to participants recruited online, participants recruited telephonically were more likely to be older on average (telephone=28.5 years versus online=25.0; $p<.001$), live in rural versus urban settings (telephone=23.0% rural versus online=4.4%; $p<.001$), and have less than college education (telephone=16.1% college-educated versus online=79.7%; $p<.001$). Half of telephone participants were male, compared to less than four in ten online participants (49.9% telephone versus 39.4% online; $p=.003$). Significant differences were also identified between recruitment methods for relationship status ($p<.001$), profession ($p<.001$), and religion ($p<.001$). Compared to telephone participants, online participants also had significantly higher COVID-19 knowledge and awareness ($p<.001$) and were significantly more likely to engage in prevention behaviors ($p<.001$).”

Limitations: “The recruitment strategy increased the potential for selection bias because participants were recruited online and on-the-ground in Senegal; thus, all participants, by the nature of the recruitment methods, had access to the internet and/or a cell phone.”

2. Throughout the paper, the introduction part is not directly related to the research purpose, various aspects of the methodology seem to be unclear, and some concern was raised about the statements in the discussion part beyond the available results.

We appreciate the reviewer’s feedback on the introduction, methodology, and discussion sections. To better relate our introduction to the study’s research purpose, we have revised the following text in the “COVID-19 in West Africa” section of the introduction (in red), including the header as shown below. We have also modified the “Study Objectives” in the introduction to clarify our study’s methodology.

Introduction:

“Senegal as a Case Study of COVID-19 Beliefs and Behaviors in West Africa

Senegal is the former capital of French West Africa and home to over 16 million people, and therefore may serve as a relevant case study for exploring COVID-19 beliefs in West Africa. Furthermore, Senegal’s communication regarding COVID-19 has widely been considered successful and may have contributed to unexpectedly fewer COVID-19 cases and deaths.[35,36]. Compared to the burden of COVID-19 in Europe, the Americas, the Middle East and South Asia, the burden of COVID-19 in Africa has been quite low. Relative to the rest of the world, Africa defied initial expectations by experiencing the lowest burden of reported COVID-19 cases and deaths.[13,14] Questions remain about why COVID-19 infections are reportedly low in Africa, in particular in West Africa where there have been 377 cases per 100,000 persons cumulatively. Although a recent analysis by Aduh and colleagues (2020) reported on perceptions and behaviors towards COVID-19 across Sub-Saharan Africa, we have much to learn from populations in densely populated West Africa, a comparatively more urban Sub-Saharan region home to more than 400 million people.[16]

Some researchers have questioned the accuracy of African COVID-19 case numbers,[13] hypothesizing that the burden of disease could be much higher than reported. Uncertainty about the true trajectory of COVID-19 in Africa may leave all continents vulnerable to future outbreaks, particularly regions with low herd immunity due to lack of exposure and vaccine resource constraints. It is likely that places that have not yet needed to adopt strong prevention efforts because of reportedly low case numbers therefore may be especially vulnerable to future outbreaks. Thus, despite relatively low case numbers, it is still vital to investigate COVID-19 health beliefs and prevention behaviors in West Africa peoples to identify potential correlates and predictors of viral prevention behaviors. Our study aimed to contribute evidence towards this gap in the literature by adapting pre-existing research panels and developing novel data collection instruments to capture information about COVID-19, and this may serve as a model for the responsiveness of ongoing scholarship to future global events.”

Introduction / Study Objectives: “To explore COVID-19 beliefs and prevention behaviors in francophone West Africa, the current study analyzes cross-sectional survey data collected from a sample of adults in Senegal. The survey, which was administered both online and via telephone, included a module on COVID-19 behaviors and beliefs. We sought to identify correlates between respondent demographic attributes and COVID-19 beliefs and behaviors.”

Other comments

Introduction

3. The authors described quite a lot of West African context (lines 6-44, page 6), but it seems not directly related to the research purpose.

Thank you for your feedback on the West African context section in our introduction. We appreciate the opportunity to revise this section to better describe why we believe Senegal is a relevant case study for exploring COVID-19 beliefs and behaviors in French West Africa. The following text has been modified in the “COVID-19 in West Africa” section of the introduction (in red), including the header as shown below.

Introduction:

“Senegal as a Case Study of COVID-19 Beliefs and Behaviors in West Africa

Senegal is the former capital of French West Africa and home to over 16 million people, and therefore may serve as a relevant case study for exploring COVID-19 beliefs in West Africa. Furthermore, Senegal’s communication regarding COVID-19 has widely been considered successful and may have contributed to unexpectedly fewer COVID-19 cases and deaths.[35,36]. Compared to the burden of COVID-19 in Europe, the Americas, the Middle East and South Asia, the burden of COVID-19 in Africa has been quite low. Relative to the rest of the world, Africa defied initial expectations by experiencing the lowest burden of reported COVID-19 cases and deaths.[13,14] Questions remain about why COVID-19 infections are reportedly low in Africa, in particular in West Africa where there have been 377 cases per 100,000 persons cumulatively. Although a recent analysis by Aduh and colleagues (2020) reported on perceptions and behaviors towards COVID-19 across Sub-Saharan Africa, we have much to learn from populations in densely populated West Africa, a comparatively more urban Sub-Saharan region home to more than 400 million people.[16]

Some researchers have questioned the accuracy of African COVID-19 case numbers,[13] hypothesizing that the burden of disease could be much higher than reported. Uncertainty about the true trajectory of COVID-19 in Africa may leave all continents vulnerable to future outbreaks, particularly regions with low herd immunity due to lack of exposure and vaccine resource constraints. It is likely that places that have not yet needed to adopt strong prevention efforts because of reportedly low case numbers therefore may be especially vulnerable to future outbreaks. Thus, despite relatively low case numbers, it is still vital to investigate COVID-19 health beliefs and

prevention behaviors in West Africa peoples to identify potential correlates and predictors of viral prevention behaviors. Our study aimed to contribute evidence towards this gap in the literature by adapting pre-existing research panels and developing novel data collection instruments to capture information about COVID-19, and this may servas a model for the responsiveness of ongoing scholarship to future global events.”

4. Line 27, page 8, a repeated “that”

Thank you for noting this typo, we have removed the extra “that” accordingly.

5. Line 54, page 8. The authors described “this study applies constructs from the Health Belief Model to answer the following questions”. However, how Health Belief Model guide the study should be described in more detail.

Thank you for your feedback on better describing how the Health Belief Model guided our study. We have modified the “Study Objectives” paragraph of the introduction by adding text about the Health Belief model that was originally in the methods section. Revisions are shown below in red.

Introduction / Study Objectives: “To explore COVID-19 beliefs and prevention behaviors in francophone West Africa, the current study analyzes cross-sectional survey data collected from a sample of adults in Senegal. The survey, which was administered both online and via telephone, included a module on COVID-19 behaviors and beliefs. We sought to identify correlates between respondent demographic attributes and COVID-19 beliefs and behaviors. Our investigation was theoretically informed from Rosenstock’s Health Belief Model (HBM),[18] a behavior change framework for public health practice positing that knowledge of a potential negative outcome and perceived threat of that outcome precede behavior change.[17] Other recent studies have similarly relied on HBM constructs to explore COVID-19. Specifically, this study applies constructs from the Health Belief Model to answer the following questions:”

Method

6. The sample size calculation should be presented.

Thank you for your feedback on sample size calculations. We did not conduct sample size power calculations because this was an exploratory cross-sectional study. We address this in our limitations section as shown below:

Limitations: “We did not identify significant relationships for many factors in our regression model because our cross-sectional study was not statistically powered to detect significant differences due.”

7. Since the measurement was self-developed based on the HBM constructs, please indicate which item belongs to which HBM construct. Please also present the Cronbach’s alpha of this scale in this study. Besides, the measurement included three scales of knowledge, belief and behavior, and the total items was only 15. It was questionable whether the measurement was comprehensive enough. We have also added

Thank you for your feedback on HBM constructs and our scaling procedures. To describe which constructs of the HBM related to items in our survey, we have added text to the “Measurement” paragraph in the methods section as shown below (in red). We have also added text to present Cronbach’s alpha to the same paragraph.

Methods / Measurement: “Survey items included a series of self-reported demographic questions (e.g., age, education) as well as a 15-item section specifically about COVID-19 (see Appendix A). Using Cronbach’s α , we assessed internal consistency (i.e., scale reliability) for all 15 COVID-19 survey ($\alpha= 0.7$; acceptable reliability). Given the novel nature of our data collection instrument, we determined the scale to be acceptably reliable. The COVID-19 section contained three measurement domains: 1) knowledge & awareness, 2) perceived threat to health, and 3) prevention behaviors. COVID-19 items were developed based on available guidance from the WHO and CDC and were reviewed by French, English, and Wolof speaking team members. Response options were binary (e.g., agree/disagree) or Likert-style (i.e., ordinal). COVID-19 response items were aggregated into distinct scales based on the three measurement domains. Each scale ranged from 0 to 15. As the COVID-19 items were novel, these “scales” were used as a proxy measure for determining the degree of knowledge and awareness, perceived risk, and practicing of known prevention behaviors including 1) handwashing, 2) using hand sanitizer, 3) wearing a mask, 4) social distancing, and 5) staying at home. Perceived COVID-19 risk items drew from the Health Belief Model including perceived susceptibility (“Are you afraid of catching COVID-19?”; “Do you think the COVID-19 epidemic poses a risk to Africans?”), perceived severity (“Do you think COVID-19 represents a danger to your health?”; “How stressed do you feel about hearing about COVID-19?”), and perceived behavioral control (“If you thought you had caught COVID-19, would you know who to contact?”).

8. For data analysis, (1) the normality of the data may needed to be tested and presented; (2) “Pearson’s correlation coefficient to explore relationships between items in the COVID-19 scales” (Lines 21-23, page 11) needed to be further clarified, as this measurement included three scales and 15 items. It was not unclear which relationship between them has been explored; (3) please clarify whether all demographic items, perceived threat, and knowledge & awareness were included in the model of regression analysis or only the ones that showed significance in bivariate analysis (regarded as potential factors of behavior) were included in the model; (4) the significant level may need to be presented.

We appreciate the review’s comments on clarifying elements of our “Data Analysis” in the methods section. We address each of the 4 comments below.

(1) the normality of the data may needed to be tested and presented

To describe normality of the data, we provide summary statistics including median and mean for each of the three scales in our Results section. We did not include a table presenting these results and noted that in the manuscript text as shown below:

Page 13: “The median knowledge score was 12.5 out of 15 (range=0 [lowest] to 15 [highest]; mean=11.8 / SD=2.6; data not presented).”

Page 13: “Median perceived threat score was 11.6 out of 15 (mean=10.9 /SD=3.5; data not presented).”

Page 14: “The median prevention behaviors score was 11 out of 15 (mean=10.5/ SD=2.3; data not presented).”

(2) “Pearson’s correlation coefficient to explore relationships between items in the COVID-19 scales” (Lines 21-23, page 11) needed to be further clarified, as this measurement included three scales and 15 items. It was not unclear which relationship between them has been explored

Thank you for your feedback. During our preliminary analysis we originally intended to use Pearson's correlation, but this was not included in our final analysis as we opted for a multivariate regression technique. The above referenced statement has therefore been removed.

(3) please clarify whether all demographic items, perceived threat, and knowledge & awareness were included in the model of regression analysis or only the ones that showed significance in bivariate analysis (regarded as potential factors of behavior) were included in the model

Thank you for the opportunity to clarify our regression modelling. All demographic items, perceived threat, and knowledge and awareness were included in the regression model. To clarify our analysis, the following text has been changed in the "Data Analysis" section of our methods:

Original: Univariate and bivariate statistics were generated to describe the sample and explore potential correlations

Revised: Descriptive statistics were generated for demographic attributes and COVID-19 items.

(4) the significant level may need to be presented.

Thank you for noting this missing information. We have added the following text to the "Data Analysis" section of our methods. We have also added the alpha level to Table 3 in our Results section.

Methods / Data Analysis: "Significance was determined at the $\alpha=0.05$ level."

Table 3 addition: "Alpha=0.05 level."

Results

9. The authors compared the differences between the participants recruited online and via telephone. What did it have to do with the research purpose? In addition, as the samples of the two recruitment methods were not from the same population group, the significant difference between the two recruitment methods might not be caused by the recruitment method but the samples themselves.

Thank you for your feedback on participant differences between recruitment strategies. Although participants were recruited via telephone or online, all participants were Senegalese adults (aged 18+). We included comparisons between the two groups to increase the transparency and rigor of our study and analysis. Because there were significant differences, we chose to control for recruitment strategy as a factor in our regression modelling. We acknowledge differences between participants in our limitations as shown below:

Limitations: "The recruitment strategy increased the potential for selection bias because participants were recruited online and on-the-ground in Senegal; thus, all participants, by the nature of the recruitment methods, had access to the internet and/or a cell phone."

10. The behavior of staying at home or social distancing may be affected by various factors such as government management measures and the working status of the participants. Therefore, it might not totally reflect the prevention behavior of someone with or without work.

Thank you for your feedback on factors that may have impacted staying at home or social distancing behaviors. While we did not control for government management measures, we did control for participants' occupations in our regression modelling. However, we did not control for where a participant worked, and have added this as a limitation to our study as shown below:

Limitations: “Staying at home or social distancing prevention behaviors may have confounded by other factors not included in our regression modelling, in particular government restrictions and whether a participant worked outside of the home or not.”

Discussion

11. The suggestion that “online communication as an important strategy to build vaccine confidence.....” seems cannot conclude directly from the result.

Thank you for your feedback. To directly tie this statement to our study findings, we have revised the paragraph in question as shown:

Original: “Our results suggest that online communication will be an important strategy to build vaccine confidence and to continue supporting prevention behaviors especially as vaccine availability increases throughout the region. Respondents from the online sample demonstrated greater knowledge and awareness of COVID-19 and engaged in prevention behaviors more often compared to respondents in the telephone sample.”

Revised: “Respondents from the online sample demonstrated greater knowledge and awareness of COVID-19 and engaged in prevention behaviors more often compared to respondents in the telephone sample, suggesting that strategies leveraging online communications and communities of users will be important to build vaccine confidence and to continue supporting prevention behaviors especially as vaccine availability increases throughout the region.”

12. The authors did not find an association between knowledge and prevention behaviors. Please elaborate on whether this result was consistent with previous studies. Please clarify whether the knowledge scale has sufficient discriminative power to detect the differences considering the difficulty as well as the small number of the items.

Thank you for your feedback to better ground our findings in the context of prior research. We have revised the paragraph referenced in our discussion section as follows (added context in red). We did not assess discriminative power of our knowledge scale, and therefore we have added this as a limitation to our study in the limitations section.

Discussion: “We did not find an association between knowledge and engaging in prevention behaviors, a finding both contrary to and in line with prior international research of infectious disease practices. For example, a 2020 study in Guyana found that knowledge was a significant predictor of practicing malaria and dengue prevention behaviors but not for Zika virus and leishmaniosis, a common protozoan-caused skin infection.[26] Specific to COVID-19, a 2020 study in China reported that participants with increased knowledge kecific to COVID-19, a 2020 study in China reported that participants with increased knowrevised the paragraph in questino notwere more likely to go to crowded places and less likely to wear masks.[27] In the current study, we found that perceived COVID-19 risk was a stronger predictor of prevention behaviors, which is also supported by previous international research.[2,28]”

VERSION 2 – REVIEW

REVIEWER	Lalla-Edward, Samanta Tresha University of the Witwatersrand Faculty of Health Sciences, Ezintsha, a division of Wits Health Consortium
REVIEW RETURNED	16-Feb-2022
GENERAL COMMENTS	bmjopen-2021-057914

	A cross-sectional study of COVID-19 knowledge, beliefs, and prevention behaviors among adults in Senegal Thank you for the opportunity to re-review your manuscript. There is significant improvement in the paper, and I now only have some minor revisions. Thank you for addressing the reviewers' comments to improve the logic and readability. Abstract: The conclusion is not related to the results. You report on prevention behaviours but your conclusions are largely on knowledge and perceived risk. The conclusion in the abstract must relate to the results of the abstract and not that of the main paper where more information would have been presented to make the conclusion relevant. Keywords: These are meant to increase the chances of your article coming up in a search therefore these should not be a repeat of words in your title. Also, there is something wrong with your keywords a from formatting perspective. Consider corona, SARS COV-2, French West Africa, something digital health Main paper Introduction: Delete '...and this may serve...events.' It seems like the authorship is tied to this statement, but it really is unclear and has no value-add to the manuscript in general. Study objectives: You have a lot of information here which are not traditional to study objective reporting. Perhaps change this to Research Questions or Study purpose Methods: Study design, participants, setting Delete 'The survey was... COVID-19.' You mention how data was collected repeatedly across the manuscript. This does need to be here as well or speak directly to the section focus. In this section you need to explain that your survey respondents were sub-groups of XX and YY studies. And move the information about the online participant description from the Data collection section up here. In this way you have a proper description of who the study participants were. Data collection
--	---

	Here and across the manuscript you refer to research panels. What does this mean: - existing research cohorts, already available data collection instruments, etc... specifically considering that a panel has another meaning in laboratory terms? Also – just to confirm, the random selection of phone numbers – these were people who consented in their main study to be contacted for other studies. If not, then (and dependent on the data privacy and security acts) it is unethical to have accessed their contact details for your survey. Is it possible to list the IRBs with reference numbers? Results: Throughout the section you say, ‘data not presented.’ Why is this the case – that you want to report on the data but not present it? Can you perhaps make a general statement in the first paragraph in the results section to explain this. From a readability perspective – use n (%) or vice versa. You now mostly have % only. Can you rather state the percentages or say majority for e.g., for 94.6 % as opposed to 9 in 10 participants. You do not need to state low and high in parentheses when you are reporting ranges. This is understood, particularly if this is not a special type of result being reported on. Discussion: Whilst improved, this section is still quite long. You have added some information but did not remove or make the original section succinct. Avoid repeating the results excessively in your discussion and conclusion – we read it in the results section, rather focus on the implications. An example of repetition: Remove (paragraph 1) ‘For example, ...behaviors.’ You say the same thing a few sentences below in the Prevention Behaviors section. Formatting/Editing: Re-read and correct for tense and grammar throughout the manuscript. The manuscript is still written using sex and gender interchangeably. You are writing about sex so keep to male and female as opposed to men and women. There are still several instances of repetition e.g., telephone, online, study taking place in Senegal. Please try and remove these. You do not need to always say ‘developing nations’ in the Discussion – just state the country. Avoid stating the obvious (which decreases the writing quality) – e.g., more men versus women, urban versus rural – if you say more men, implicit is the comparator being women, if you urban implicit is the comparator urban. E.g. men may also be more likely to engage in risk taking behaviors,[26] whereas women are more risk averse.[27] most participants completed the survey via phone rather than online, respectively 84.1% and 15.9%. You reported earlier
--	--

	telephone=1,221; online = 231 so more phone than online is already known.
--	---

VERSION 2 – AUTHOR RESPONSE

Reviewer: 2

Dr. Samanta Tresha Lalla-Edward, University of the Witwatersrand Faculty of Health Sciences
Comments to the Author:

Thank you for submitting a revised manuscript. The revisions have improved the quality of the paper. Please see my minor revision comments in the attached document.

Reviewer: 2

Competing interests of Reviewer: No competing interests to declare.

1 bmjopen-2021-057914 A cross-sectional study of COVID-19 knowledge, beliefs, and prevention behaviors among adults in Senegal Thank you for the opportunity to re-review your manuscript. There is significant improvement in the paper, and I now only have some minor revisions. Thank you for addressing the reviewers' comments to improve the logic and readability.

Abstract:

- The conclusion is not related to the results. You report on prevention behaviours but your conclusions are largely on knowledge and perceived risk. The conclusion in the abstract must relate to the results of the abstract and not that of the main paper where more information would have been presented to make the conclusion relevant.

We thank the reviewer for their comments on the abstract results and conclusions sections. We have made the following changes to focus the conclusion on the results presented in the abstract (revisions in red):

Original:

“Results: Mask wearing, hand washing, and use of hand sanitizer were most frequently reported. Social distancing and staying at home were also reported albeit to a lower degree. We also identified a range of psychosocial and demographic predictors for COVID-19 prevention behaviors. Men, compared to women, had lower odds (AOR=0.59, 95% CI: 0.46-0.75, $p<.001$) of reporting prevention

behaviors. Rural residents (vs. urban; AOR=1.49, 95% CI: 1.12-1.98, p=.001) and participants with at least a high school education (vs. less than high school education; AOR=1.33, 95% CI: 1.01-1.76, p=.006) were more likely to report COVID-19 prevention behaviors.

Conclusions: In Senegal, we observed that knowledge and perceived risk of COVID-19 were very high in general. But, risk was a stronger and more influential predictor of COVID-19 prevention behaviors. Stakeholders and decision makers in Senegal and across Africa can use place-based evidence like ours to address COVID-19 risk factors and intervene effectively with policies and programming. Use of both phone and online surveys enhances representation and study generalizability and should be considered in future research with hard-to-reach populations.”

Revised:

“Results: Mask wearing, hand washing, and use of hand sanitizer were most frequently reported. Social distancing and staying at home were also reported albeit to a lower degree. Knowledge and perceived risk of COVID-19 were very high in general, but risk was a stronger and more influential predictor of COVID-19 prevention behaviors. Men, compared to women, had lower odds (AOR=0.59, 95% CI: 0.46-0.75, p<.001) of reporting prevention behaviors. Rural residents (vs. urban; AOR=1.49, 95% CI: 1.12-1.98, p=.001) and participants with at least a high school education (vs. less than high school education; AOR=1.33, 95% CI: 1.01-1.76, p=.006) were more likely to report COVID-19 prevention behaviors.

Conclusions: In Senegal, we observed high compliance with recommended COVID-19 prevention behaviors among our sample of respondents, in particular for masking and personal hygiene practice. We also identified a range of psychosocial and demographic predictors for COVID-19 prevention behaviors such as knowledge and perceived risk. Stakeholders and decision makers in Senegal and across Africa can use place-based evidence like ours to address COVID-19 risk factors and intervene effectively with policies and programming. Use of both phone and online surveys enhances representation and study generalizability and should be considered in future research with hard-to-reach populations.”

Keywords:

- These are meant to increase the chances of your article coming up in a search therefore these should not be a repeat of words in your title. Also, there is something wrong with your keywords a from formatting perspective. Consider corona, SARS COV-2, French West Africa, something digital health

We appreciate the reviewer’s feedback on our manuscript keywords. As these keywords were selected from a predetermined list provided by the publisher and authors are not permitted to create their own keywords, we believe these are appropriate keywords that accurately capture the content of our manuscript within the constraints of the currently available keyword list.

Main paper Introduction:

- Delete ‘...and this may serve...events.’ It seems like the authorship is tied to this statement, but it really is unclear and has no value-add to the manuscript in general.

We thank the reviewer for their feedback and have removed the statement identified above from the Introduction section.

Study objectives:

- You have a lot of information here which are not traditional to study objective reporting. Perhaps change this to Research Questions or Study purpose

We appreciate the reviewer’s feedback on clarifying the above heading in our Introduction section. The heading has been changed from “Study Objectives” to “Study Objectives and Research Questions”.

Methods: Study design, participants, setting

- Delete ‘The survey was... COVID-19.’ You mention how data was collected repeatedly across the manuscript. This does need to be here as well or speak directly to the section focus.

We thank the reviewer for their feedback on reducing redundant text in our Methods section, and have removed the above sentence accordingly.

- In this section you need to explain that your survey respondents were sub-groups of XX and YY studies. And move the information about the online participant description from the Data collection section up here. In this way you have a proper description of who the study participants were.

We thank the reviewer for their feedback on our Methods section. Survey respondents for the telephone sample were not sub-groups but were all part of a pre-existing cohort of survey respondents for a larger cohort study. As stated in the text, online participants were also part of a pre-existing cohort of survey respondents for a larger cohort study; however, only online participants from Senegal were included and cohort participants from other countries were excluded from the current analysis. This information is also presented in Figure 1, a flow diagram of our study recruitment and sampling. To clarify this point in the first paragraph of our methods section, the below sentence was revised (changes in red):

“All respondents were participating in a separate ongoing longitudinal cohort study.”

We appreciate the reviewer's feedback on providing a proper description of who the study participants were, and have revised the "Study Design, Participants and Setting" and "Data Collection" sections accordingly as shown below.

Original:

"Study Design, Participants and Setting

From June-August 2020, we conducted a multi-modal cross-sectional survey that included questions regarding COVID-19 beliefs and prevention behaviors among a convenience sample of 1,452 adults in Senegal. The survey was administered via the internet and telephone, which was appropriate given the constraints of research during COVID-19. Respondents were participating in a separate ongoing longitudinal cohort study.

Data Collection

This survey used two modes of data collection: online and telephone. We opted to collect data about COVID-19 from pre-existing panels of participants in Senegal so as to be responsive to the ongoing and evolving global epidemic and to address evidence gaps mentioned above (see Introduction). The telephone survey used a random selection from a panel of phone numbers of individuals who lived in Senegal and agreed to participate in media-related research. A quota system was used to match the overall population of Senegal on sex and have an even distribution of age and level of education. A total of 1,221 participants were enrolled in the study via telephone, split approximately equally between participants living inside versus outside of Dakar, Senegal and by gender. Online participants from Senegal were recruited using social media advertisements on Facebook and YouTube. All online participants were part of a cohort study about entertainment education in West Africa that began in late summer 2019; of the 1,286 cohort participants recruited online in summer 2019, 231 both were eligible to participate (i.e., Senegalese adults) and completed the COVID-19 questionnaire items. Through our multi-modal strategy, responses about COVID-19 were collected from 1,452 Senegalese adults. Figure 1 presents a flow diagram of our study recruitment and sampling.

For both online and telephone participants, all individuals were required to be 18 years or older. Potential participants were informed about the study and provided consent to participate in the survey (online) or interview (telephone). Consenting participants were compensated with 2000 Central African CFA franc (~4 USD) for their time. Telephone surveys were collected by OMedia, a Dakar-based third party research firm, and overseen by researchers from UCLA (MB, DG). Online surveys were collected via Qualtrics by researchers from Drexel University (MK, PM). Incomplete responses with missing data were excluded. All study procedures and protocols were reviewed by and approved or determined to be exempt by institutional ethical review boards."

Revised:

"Study Design, Participants and Setting

From June-August 2020, we conducted a multi-modal cross-sectional survey that included questions regarding COVID-19 beliefs and prevention behaviors among a convenience sample of 1,452 adults in Senegal. All respondents were participating in a separate ongoing longitudinal cohort study. Potential participants were informed about the study and provided consent to participate in the survey (online) or interview (telephone). For both online and telephone participants, all individuals were required to be 18 years or older.

All online participants were part of a cohort study about entertainment education in West Africa that began in late summer 2019; of the 1,286 cohort participants recruited online in summer 2019, 231 both were eligible to participate (i.e., Senegalese adults) and completed the COVID-19 questionnaire items. Online participants from Senegal were recruited using social media advertisements on Facebook and YouTube.

The telephone survey used a random selection of phone numbers from a pre-existing research cohort of individuals who lived in Senegal and agreed to participate in media-related research and subsequent follow up surveys. COVID-19 questionnaire items were included as an accompanying section with the primary study's survey. A quota system was used to match the overall population of Senegal on sex and have an even distribution of age and level of education. A total of 1,221 participants were enrolled in the study via telephone, split approximately equally between participants living insider versus outside of Dakar, Senegal and by gender.

Data Collection

This survey used two modes of data collection: online and telephone. We opted to collect data about COVID-19 from pre-existing cohorts of participants in Senegal so as to be responsive to the ongoing and evolving global epidemic and to address evidence gaps mentioned above (see Introduction). Through our multi-modal strategy, responses about COVID-19 were collected from 1,452 Senegalese adults. Consenting participants were compensated with 2000 Central African CFA franc (~4 USD) for their time. Telephone surveys were collected by OMedia, a Dakar-based third party research firm, and overseen by researchers from UCLA (MB, DG). Online surveys were collected via Qualtrics by researchers from Drexel University (MK, PM). Incomplete responses with missing data were excluded. All study procedures and protocols were reviewed by and approved or determined to be exempt by institutional ethical review boards (see Research Ethics Approval). Figure 1 presents a flow diagram of our study recruitment and sampling.”

Data collection

- Here and across the manuscript you refer to research panels. What does this mean: - existing research cohorts, already available data collection instruments, etc... specifically considering that a panel has another meaning in laboratory terms?

We thank the reviewer for the opportunity to clarify the language in our survey. We have removed all references to research panels and instead adopted the widely-used phrase “research cohorts”.

- Also – just to confirm, the random selection of phone numbers – these were people who consented in their main study to be contacted for other studies. If not, then (and dependent on

the data privacy and security acts) it is unethical to have accessed their contact details for your survey.

We appreciate the opportunity to clarify the text in our methods section regarding the telephone research cohort participants. We have added the following text to clarify that participants' privacy was not violated and that the COVID-19 questions were asked simultaneously with questions pertaining to the main research study (additions in red):

“The telephone survey used a random selection of phone numbers from a pre-existing research cohort of individuals who lived in Senegal and agreed to participate in media-related research and subsequent follow up surveys. COVID-19 questionnaire items were included as an accompanying section with the primary study's survey.”

- Is it possible to list the IRBs with reference numbers?

We appreciate the opportunity to provide additional details regarding our ethical approval. We have added the IRB reference number for the second institutional review board to the “Research Ethics Approval” section preceding the acknowledgments of our manuscript, and have referenced this in our methods section as indicated below (additions in red):

“All study procedures and protocols were reviewed by and approved or determined to be exempt by institutional ethical review boards (see Research Ethics Approval).”

Results:

- Throughout the section you say, ‘data not presented.’ Why is this the case – that you want to report on the data but not present it? Can you perhaps make a general statement in the first paragraph in the results section to explain this.
- From a readability perspective – use n (%) or vice versa. You now mostly have % only.
- Can you rather state the percentages or say majority for e.g., for 94.6 % as opposed to 9 in 10 participants.
- You do not need to state low and high in parentheses when you are reporting ranges. This is understood, particularly if this is not a special type of result being reported on.

We thank the reviewer for the suggested revisions to our Results section. To address the reviewer's comment about the phrase “data not presented”, we have added the below text (in red) to the first paragraph of our Results section to indicate to readers that we are happy to provide data not presented upon request. Regarding the reviewer's recommendation to improve readability by including frequencies as well as percentages, we have added in frequencies in addition to percentages. While we appreciate the reviewer's suggestion to avoid using terms like “nine in ten”, we

believe that use of such terms in prose is appropriate because presenting data in multiple formats increases the accessibility of our manuscript to larger audiences, hence our decision to also include percentages plus frequencies to provide the fullest context to readers. Finally, we removed references to “low” and “high” in accordance with the reviewer’s feedback.

“A total of 1,452 Senegalese adults participated in our cross-sectional survey about COVID-19. Table 1 presents demographics characteristics for the study sample. Table 2 presents univariate descriptive statistics for COVID-19 survey items. Table 3 presents unadjusted regression models and Table 4 presents adjusted regression models. Figure 2 presents a Forest plot of our logistic regression model predicting prevention behaviors. Note: To provide context and in light of space limitations, we report on some findings without presenting related data which may be furnished upon reader request.”

Discussion:

- Whilst improved, this section is still quite long. You have added some information but did not remove or make the original section succinct. Avoid repeating the results excessively in your discussion and conclusion – we read it in the results section, rather focus on the implications.
- An example of repetition: Remove (paragraph 1) ‘For example, ...behaviors.’ You say the same thing a few sentences below in the Prevention Behaviors section.

We appreciate the reviewer’s feedback on our discussion section. We have removed the requested sentence from our first paragraph and reviewed the remaining sections for redundant language related to our results accordingly.